# FitLight: Federated Imitation Learning for Plug-and-Play Traffic Signal Control

## Abstract

Although Reinforcement Learning (RL)-based Traffic Signal Control (TSC) methods have been extensively studied, their practical applications still raise some serious issues, such as high learning cost and poor generalizability. This is because the "trial-and-error" training style makes RL agents extremely dependent on the specific traffic environment, which also requires a long convergence time. To address these issues, we propose a novel federated imitation learning-based framework for multi-intersection TSC, named FitLight, which allows RL agents to plug-and-play for any traffic environment without additional pre-training cost. Unlike existing imitation learning approaches that rely on pre-training RL agents with demonstrations, FitLight allows real-time imitation learning and seamless transition to reinforcement learning. Due to our proposed knowledge-sharing mechanism and novel hybrid pressure-based agent design, RL agents can quickly find a best control policy with only a few episodes. Moreover, for resource-constrained TSC scenarios, FitLight supports model pruning and heterogeneous model aggregation, such that RL agents can work on a microcontroller with merely $16KB$ RAM and $32KB$ ROM. Extensive experiments demonstrate that, compared to state-of-the-art methods, FitLight not only provides a superior starting point but also converges to a better final solution on both real-world and synthetic datasets, even under extreme resource limitations.

## 1 Introduction

With the rapid development of cities and the rapid growth of population, an increasing number of cities are suffering from severe traffic congestion, resulting in various serious problems, including economic losses, rising commuting costs, and environmental pollution. The *Traffic Signal Control* (TSC) has attracted widespread attention as a promising solution to traffic congestion (Chen et al., 2013; Waszecki et al., 2017; Chang et al., 2020). In most real-world applications, the control policies are rule-based (e.g., FixedTime (Koonce & Rodegerdts, 2008), GreenWave (Török & Kertész, 1996), SCOOT (Hunt et al., 1982), and SCATS (PR, 1992)) that follow some pre-defined rules of the traffic plan. To better handle dynamic traffic scenarios, several adaptive methods have been proposed (e.g., MaxPressure (Varaiya, 2013), MaxQueue (Zhang et al., 2021), and SOTL (Cools et al., 2013)), which control traffic in a heuristic manner. Due to the advancement of Artificial Intelligence (AI) and Internet of Things (IoT) technologies, utilizing Reinforcement Learning (RL) to control traffic signals has been a promising approach.

Although RL-based methods can achieve better control performance, their usage is greatly restricted by the issues of high learning cost and poor generalizability. This is due to the trial-and-error learning style of RL agents, which requires the RL agent to make a large number of attempts in a specific traffic environment to gradually learn the control strategy. Worse still, as the size of the road network and the number of RL agents increase, the size of the policy space will also increase exponentially, making it extremely difficult to find the optimal control strategy in multi-intersection scenarios and requiring substantial training costs. Therefore, *how to effectively improve both the training efficiency and generalization ability is becoming a major challenge in designing RL-based TSC methods.*

To tackle this problem, in this paper, we propose a novel Federated Imitation Learning (FIL)-based framework named FitLight for efficient and effective multi-intersection TSC, which enables RL agents to plug-and-play for different traffic environments without additional pre-training cost. In other words, after obtaining a very high-quality solution in the first episode, FitLight can quickly converge to a better final control strategy. As shown in Figure 1, FitLight is built on a cloud-edge framework consisting of one cloud server and multiple edge nodes (i.e., RL agent and its

corresponding intersection). Unlike existing methods that either train RL agents directly within the traffic environment or employ imitation learning over pre-collected data for pre-training, FitLight seamlessly integrates imitation learning into the reinforcement learning process. This integration enables the RL agent to achieve a high-quality initial solution in the first episode, thanks to the supervision provided by imitation learning. Subsequently, due to our novel hybrid pressure-based agent design, the RL agent seamlessly transitions into the reinforcement learning phase, ultimately converging to an even better control strategy. Moreover, FitLight's support for model pruning and heterogeneous model aggregation ensures that RL agents can be deployed in resource-constrained TSC scenarios. In summary, this paper makes the following four major contributions:

- We propose a novel Federated Imitation Learning (FIL)-based framework that can plug and play for different traffic environment without incurring additional pre-training costs.

- We introduce an imitation learning mechanism combined with a hybrid pressure-based agent design, enabling real-time imitation learning and smooth transitions to reinforcement learning, allowing RL agents to quickly achieve a high-quality solution in the first episode.

- We propose a federated learning-based knowledge sharing mechanism that supports heterogeneous model aggregation, thereby improving learning efficiency and enabling our FitLight to operate in TSC scenarios with extremely limited resources.

- Extensive experiments on various synthetic and real-world datasets show the superiority of our FitLight in terms of average travel time and convergence rate.

## 2 RELATED WORK

To improve the performance of TSC, various methods based on RL have been proposed. For example, PressLight (Wei et al., 2019a), CoLight (Wei et al., 2019b), MPLight (Chen et al., 2020), MetaLight (Zang et al., 2020), and RTLight (Ye et al., 2023a) performed TSC optimization based on the concept of pressure from the Max Pressure (MP) control theory (Varaiya, 2013) to design the state and reward. Unlike

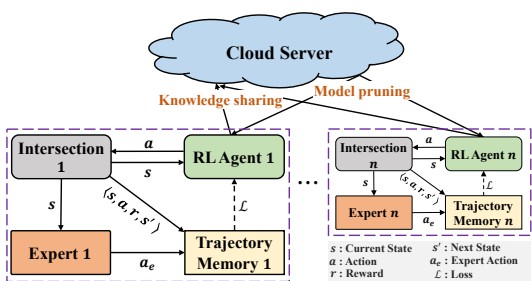

Figure 1: Framework of FitLight.

these methods, IPDALight (Zhao et al., 2022) proposed a new concept named intensity, which investigates both the speed of vehicles and the influence of neighboring intersections. To reflect the fairness of individual vehicles, FairLight (Ye et al., 2022) and FELight (Du et al., 2024) considered the relationship between waiting time and driving time, as well as the extra waiting time of vehicles, in a deceptive manner. To leverage cooperation among RL agents in the road network, FedLight (Ye et al., 2021) and RTLight (Ye et al., 2023a) utilize federated reinforcement learning to share knowledge. HiLight (Xu et al., 2021) cooperatively controls traffic signals to directly optimize average travel time by using hierarchical reinforcement learning. UniLight (Jiang et al., 2022) uses a universal communication form between intersections to implement cooperation. However, the RL agents of these methods are trained from randomly initialized models, resulting in a long training time before obtaining the final control strategy.

To improve learning efficiency, imitation learning (Agarwal et al., 2019; Argall et al., 2009; Hussein et al., 2017; Osa et al., 2018) that enables the RL agent to learn from expert demonstrations is a promising approach. Currently, imitation learning can be divided into two categories: behavioral cloning (Bain & Sammut, 1995; Pomerleau, 1991) and adversarial imitation learning (Abbeel & Ng, 2004; Syed & Schapire, 2007; Ziebart et al., 2008), both of which have been applied in TSC. Specifically, DemoLight (Xiong et al., 2019) is a behavioral cloning-based method that reduces the imitation learning task to a common classification task (Ross & Bagnell, 2010; Syed & Schapire, 2010) by minimizing the action difference between the agent strategy and the expert strategy. However, since this method is trained in a single-intersection environment and relies on pre-collected expert trajectories from the same environment, it cannot be applied to multi-intersection scenarios and is highly specific to the training environment. On the other hand, as an adversarial imitation learning-based method, InitLight (Ye et al., 2023b) uses a generative adversarial framework to learn experts' behaviors, where the discriminator iteratively differentiates between pre-collected expert and agent trajectories (generated through real-time agent-environment interactions). Although InitLight can use trajectories from different environments to train RL agents, it still needs a pre-training process to obtain an initial model.

Figure 2: Architecture and workflow of FitLight.

To the best of our knowledge, FitLight is the first federated imitation learning framework for TSC, enabling RL agents to plug-and-play in any traffic environment without additional pre-training costs. This allows the RL agent to achieve a high-quality initial solution in the first episode and then converge to an even better final result.

## 3 OUR FITLIGHT APPROACH

To enable RL agents to plug-and-play in various traffic scenarios without pre-training, we design a novel FitLight approach based on a cloud-edge architecture, where the cloud server facilitates knowledge sharing among intersections. Each intersection is equipped with an RL agent and an expert strategy. Figure 2 details the FitLight components and workflow. In our approach, the federated imitation learning framework comprises a cloud server and multiple edge nodes. The cloud server first dispatches pruned models from a base model to edge nodes and then shares knowledge among different intersections by aggregating heterogeneous models during the RL training. For each edge node, we deploy an RL agent to monitor traffic dynamics using connected sensors, such as cameras, to make traffic signal control decisions and update network parameters. Once capturing the current traffic state $s$, the RL agent will choose one best action $a$ to control traffic lights. Meanwhile, the expert strategy also gives a decision $a_e$, which will be stored as the label of the current state for imitation learning. This expert guidance enables the RL agent to quickly identify a high-quality solution. We will give the details of our approach in the following subsections.

### 3.1 INTERSECTION MODELING

The right part of Figure 2 shows an intersection example with three components, i.e., arrival and departure lanes, directed roads, and control phase setting:

- **Arrival and Departure Lanes:** The intersection consists of a set of arrival lanes $L_a = \{l_1, l_2, \cdots, l_{12}\}$ and a set of departure lanes $L_d = \{l'_1, l'_2, \cdots, l'_{12}\}$, where vehicles can enter and exit the intersection, respectively.

- **Directed Roads:** Based on direction marks at the end of each arrival lane, we define a directed road as $(l_a, l_d), l_a \in L_a, l_d \in L_d$, where $l_d$ is the departure lane indicated by the direction mark on the ground of $l_a$. For example, $(l_7, l'_7)$ and $(l_8, l'_8)$ are two directed roads.

- **Control Phases:** According to common sense, the vehicles turning right are not restricted by traffic. Therefore, we design a set of eight feasible control phases $P = \{p_1, p_2, \cdots, p_8\}$, which are obtained by combining 2 of 8 directed roads and indicate the rights-of-way signaled to vehicles by traffic lights. For example, the intersection on the right side of Figure 2 shows a scenario with control phase $p_4$ enabled, where the vehicles on lane $l_7$ can turn left to enter lane $l'_7$ and the vehicles on lane $l_8$ can go straight to enter lane $l'_8$. Note that the number of control phases is fixed, regardless of the number of lanes.

### 3.2 HYBRID PRESSURE

Unlike many existing works that use pressure from MP control theory to model traffic dynamics, this paper introduces a novel concept called Hybrid Pressure (HP) for designing RL agents. Specifically, HP takes into account a broader range of dynamics, from individual vehicle behavior to intersection-level interactions, rather than solely focusing on the number of vehicles, as is common in MP. This approach allows for more accurate modeling of RL elements.

**Definition 1** *(Hybrid Pressure of a Vehicle). The hybrid pressure of a vehicle $veh$ on a lane of an intersection $hp_{veh}$ is defined as:*

$$hp_{veh} = log(1 + \frac{l_{max} - d}{l_{max}} + \frac{v_{max} - v}{v_{max}} + \frac{wt_{veh}}{dt_{veh}}), \quad (1)$$

*where $l_{max}$ is the lane length, $d$ indicates the distance between the vehicle and the intersection, $v_{max}$ is the maximum speed of the lane, $v$ is the current speed of the vehicle, $wt_{veh}$ and $dt_{veh}$ are the overall waiting time and driving time of the vehicle along its route so far from the time when it entering the traffic network, respectively. Note that we normalize the distance and speed by $l_{max}$ and $v_{max}$ to constrain their ranges. Moreover, we use $log(\cdot)$ to smooth the absolute value of $h_{veh}$ and plus 1 to make sure $h_{veh}$ is always greater than 0.*

We use the newly designed hybrid pressure of a vehicle to indicate the traffic priority when the vehicle arrives at an intersection. According to Definition 1, we can find that the vehicles with a shorter distance to the intersection, a slower speed, and a longer cumulative waiting time will have a higher $h_{veh}$ value, i.e., have greater priority for the right of way. When waiting at an intersection, the waiting time of a vehicle increases cumulatively, resulting in an increase in the corresponding lane's HP value. Along with the increasing lane HP values, the vehicles on some feeder roads can eventually move. Due to the elegant combination of individual vehicles' features, HP can more accurately reflect the traffic dynamics.

**Definition 2** *(Hybrid Pressure of a Directed Road). The hybrid pressure of a directed road $(l_a, l_d)$ is defined as the difference in hybrid pressure between the arrival and departure lanes, where a lane's hybrid pressure is the sum of all vehicles' HP on that lane.*

$$hp_{(l_a, l_d)} = \sum_{veh \in l_a} hp_{veh} - \sum_{veh \in l_d} hp_{veh}. \tag{2}$$

To denote the hybrid pressure of all the vehicles on a lane, Definition 2 defines the HP of a directed road $(l_a, l_d)$. Since the hybrid pressure of a lane is in a summation form, it implicitly reflects the number of vehicles on both $l_a$ and $l_d$. Therefore, $hp_{(l_a, l_d)}$ not only reflects the status of individual vehicles but also captures the imbalance in traffic conditions between the upstream and downstream lanes. In our approach, the TSC controller tends to allow the directed road with higher HP values to move first.

**Definition 3** *(Hybrid Pressure of an Intersection). The hybrid pressure of an intersection I equals the difference between the arrival and departure lanes' HP values, i.e.,*

$$hp_I = \sum_{l \in L_a} hp_l - \sum_{l \in L_d} hp_l, \tag{3}$$

Definition 3 presents how to calculate the FI for an intersection, which can be used to evaluate the overall traffic pressure faced by the intersection. From this definition, we can see that the hybrid pressure of the intersection can approximately reflect the imbalance between upstream and downstream traffic statuses at the intersection. Therefore, similar to the MP control theory, if the HP of the intersection can be controlled at a low level, the throughput of vehicles crossing this intersection will be maximized.

### 3.3 Edge Node Design

In our approach, the traffic lights of each edge node are controlled by a Proximal Policy Optimization (PPO) (Schulman et al., 2017) agent. Unlike most existing works that directly train the agent through reinforcement learning, we deploy an expert algorithm to guide the agent towards efficient convergence via imitation learning, enabling the RL agent to find a high-quality solution in the first episode.

**Expert Algorithm.** Based on the concept of hybrid pressure, we design a simple but effective control heuristic named MaxHP, which greedily selects the control phase with the maximum HP values. Similar to the classic MP-based heuristic control method MaxPressure (Varaiya, 2013), by allowing vehicles in the lane with the largest HP value to pass, MaxHP can reduce the HP value of the intersection.

**Agent Design.** In this paper, we design the key elements of the PPO agent by using the proposed HP concept, i.e.,

- **State:** State is the information of the intersection captured by the agent as its own observation for phase selection. Take the standard intersection in Figure 2 as an example, the state includes the HP of all directed lanes (i.e., $hp_{l_1, l'_1}, hp_{l_2, l'_2}, \cdots, hp_{l_{12}, l'_{12}}$ and the current control phase (i.e., $p_4$)).

- **Action:** Based on the observed current traffic state, the PPO agent needs to choose one best control phase to maximize the throughput of the intersection. For the intersection example in Figure 2, the PPO agent has 8 permissible control phases (i.e., $p_1, \cdots, p_8$).

- **Reward:** Once an action is completed, the environment will return a reward to the agent. The reward mechanism plays an important role in the RL learning process. It is required that a higher reward implies a better action choice. As mentioned in Definition 3, to encourage the agent to maximize the throughput of the intersection by minimizing the hybrid pressure of the intersection $hp_I$, in this paper, we define the reward as $r = -hp_I$.

A PPO agent consists of two trainable networks, i.e., Actor $\theta_A$ and Critic $\theta_C$, where $\theta$ is the model parameter. The Actor model is responsible for learning the policy and determining which action to take given the current state. The Critic model, on the other hand, serves as a value estimator, assessing whether the action selected by the Actor will lead to an improved state in the traffic environment. Therefore, the feedback from the Critic model can also be used to optimize the Actor model. In this paper, since we also utilize imitation learning to guide the agent training, as shown in the right part of Figure 2, there are two loss functions from reinforcement learning $L_R$ and imitation learning $L_I$. To calculate these loss values for optimizations, a mini-batch of trajectory samples is collected from the agent trajectory memory, where each sample is a quintuple $\langle s, a, a_e, r, s' \rangle$.

First, the reinforcement learning loss $L_R$ includes the losses of both the Critic model $L_C$ and the Actor model $L_A$. Note that the reinforcement learning of PPO requires that trajectory samples in a mini-batch be continuous.

_Critic Model._ We optimize the Critic model by:

$$L_C = \mathbb{E}[|\theta_C(s_t)_{target} - \theta_C(s_t)|], \tag{4}$$

where $\mathbb{E}$ is an operator to calculate the empirical average over a mini-batch of samples, and $\theta_C(s_t)_{target}$ can be calculated as $\theta_C(s_t)_{target} = r_{t+1} + \gamma \cdot C(s_{t+1})$ by using the Temporal-Difference (TD) algorithm (Tesauro et al., 1995) to estimate the target value.

_Actor Model._ As a policy gradient-based RL algorithm, the objective of the Actor model is

$$L_A = \mathbb{E}[min(R_t, clip(R_t, 1 - \sigma, 1 + \sigma))A_t], \tag{5}$$

where $R_t = \frac{\theta_A(a_t|s_t)}{\theta_A^{old}(a_t|s_t)}$ is the importance sampling that obtains the expectation of samples under the new Actor model $\theta_A$ we need to update, $A_t$ is an estimated value of the advantage function at time step $t$, and $\sigma$ is the clipping parameter that restricts the upper/lower bounds in the $clip(\cdot)$ function to stabilize the updating process. Note that the samples are gathered from an old Actor model $\theta_A^{old}$. The advantage function $A_t$ is computed with the Generalized Advantage Estimator (GAE) (Schulman et al., 2015) as

$$\begin{aligned} A_t &= \delta_t + (\gamma\lambda)\delta_{t+1} + (\gamma\lambda)^2\delta_{t+2} \\ &+ \cdots + (\gamma\lambda)^{|B|-t+1}\delta_{|B|-1}, \end{aligned} \tag{6}$$

where $\gamma \in [0, 1]$ is the discount factor of future rewards, $\lambda \in [0, 1]$ is the GAE parameter, $|B|$ is the batch size of the sampled mini-batch, and $\delta_t = r_t + \gamma\theta_C(s_{t+1}) - \theta_C(s_t)$.

In addition to the reinforcement learning loss, the Actor model also has a loss from imitation learning, which guides the agent's behavior.

_Imitation Learning._ In our approach, we use MaxHP as the expert algorithm to label the state $s$ by selecting the corresponding action $a_e$. This labeling process enables us to apply supervised learning to minimize the discrepancy between the Actor's actions and the expert's actions. Since the control phases are discrete actions, we employ the cross-entropy loss function (Shannon, 1948; Zhang & Sabuncu, 2018) to handle this multi-class classification task:

$$L_I = -\sum_{i=1}^{|P|} a_{e_i} log(a_i), \tag{7}$$

where $|P|$ is the number of classes (control phases), $a_{e_i}$ is the indicator variable of the label (i.e., the action chosen by the expert algorithm) that is encoded by a one-hot vector, $a_i$ is the prediction probability of the $i$-th action given by the Actor model.

Finally, considering the balance of exploitation and exploration for the RL agent training, we use a balance factor $\alpha$ to adjust the weighting of different losses:

$$L = \alpha(L_C + L_A) + (1 - \alpha)L_I, \tag{8}$$

where $\alpha$ increases with the number of training episodes, facilitating a gradual transition from imitation learning to reinforcement learning.

### 3.4 CLOUD SERVER DESIGN

In our approach, we use a cloud server to co-ordinate the training process among RL agents at different intersections. Specifically, for traffic scenarios with extremely limited resources, the cloud server first sends pruned initial models that meet the requirements to each intersection. During the training process, the cloud server facilitates knowledge sharing by aggregating gradient information from these heterogeneous models, enabling effective collaboration among agents at different intersections.

**Model Pruning.** To meet resource requirements, we employ structured pruning at initialization. This technique leverages the concept that a randomly initialized dense network contains a subnetwork (referred to as a "winning ticket") capable of achieving performance comparable to the original dense network (Frankle & Carbin, 2018; Kim & Sung, 2023). Specifically, for a dense network with parameters $\theta$, network pruning results in a new model $\theta \odot M$, where $M = \{0, 1\}^{|\theta|}$ is a binary mask used for the pruning, and $\odot$ denotes the Hadamard product (element-wise multiplication). In our approach, we generate multiple Actor and Critic subnetworks for each intersection, applying a fixed set of pruning ratios to different network layers. As illustrated in the left part of Figure 2, we create three pruned submodels from the base model for different intersections. In these submodels, lighter colors indicate pruned neurons, while darker colors represent retained neurons.

---

**Algorithm 1: FitLight Training Procedure**

1 **Input:** episodes $S$; episode steps $T$; trajectory memory $M_T$; actor model $\theta_A$; critic model $\theta_C$; expert $E$; batch size $B$.
2 **Output:** trained models $\theta_C$ and $\theta_A$.
3 receive model structures from cloud server;
4 randomly initialize $\theta_A$ and $\theta_C$;
5 **for** $episode = 1$ **to** $S$ **do**
6    **for** $step = 1$ **to** $T$ **do**
7       obtain current intersection traffic state $s$;
8       choose action $a$ based on $s$;
9       obtain expert behavior $a_e$ from $E$;
10      execute action $a$ at the intersection;
11      observe next state $s'$ of the intersection;
12      store trajectory $\langle s, a, a_e, r, s' \rangle$ in $M_T$;
13      **if** $Size(M_T) \geq B$ **then**
14         sample mini-batch $b$ from $M_T$;
15         compute $L_C$ by Equation 4;
16         compute $L_A$ by Equation 5;
17         compute $L_I$ by Equation 7;
18         compute $L$ by Equation 8;
19         update model $\theta_C$ and $\theta_A$;
20         upload gradient $\nabla L$ to cloud server for aggregation (Eq. 9);
21         receive $\overline{\nabla L}$ from cloud server to update $\theta_C, \theta_A$;
22      **end**
23    **end**
24 **end**
25 **return** $\theta_C, \theta_A$

---

**Knowledge Sharing.** As a specialized form of supervised learning, imitation learning also requires a substantial number of samples to train RL agents effectively. To enhance learning efficiency and maximize the use of trajectory samples, we introduce a knowledge-sharing mechanism that aggregates gradients from heterogeneous submodels. Specifically, since the submodels for each intersection are derived from the same base model, we aggregate their gradients using a weighted average operation as follows:

$$\overline{\nabla L} = \frac{\sum_{i=1}^{N} \nabla L_i}{\sum_{i=1}^{N} M_i}, \tag{9}$$

where $N$ is the number of intersections, $\nabla L_i$ and $M_i$ represent the gradient of the loss function and the binary mask from the $i$-th intersection's submodel, respectively.

As shown in the left part of Figure 2, each colored neural network represents different agents' subnetworks, where the generated subnetworks share some subsets of parameters across multiple agents. In the aggregated model, each neuron is colored according to the colors of agents that share the corresponding neuron.

### 3.5 FITLIGHT IMPLEMENTATION

Algorithm 1 details the training process of a FitLight agent. In lines 1-2, the agent is initialized with the pruned structure. Lines 5-10 illustrate the interaction between the agent and the intersection, where the exact algorithm assigns the label $a_e$ to the current state $s$. When the agent stores enough trajectory samples, lines 12-16 update the parameters of the PPO agent by using the local loss. Lines

18-19 show our knowledge-sharing mechanism, where the cloud server collects the gradient from all intersections and then dispatches the aggregated gradient to them for parameter update.

## 4    PERFORMANCE EVALUATION

To evaluate the effectiveness of our approach, we conduct experiments on a Ubuntu server equipped with an Intel Core i9-12900K CPU, 128GB of memory, and an NVIDIA RTX 3090 GPU. We implement our FitLight approach using the open-source traffic simulator Cityflow (Zhang et al., 2019) in Python. During the simulation of traffic scenarios, similar to prior work (Wei et al., 2019a; Ye et al., 2021; 2023b), we set the phase duration to 10 seconds. In FitLight, the PPO agent comprises two neural networks: the Actor model, which has three layers (containing 13, 32, and 8 neurons), and the Critic model, which has three layers (containing 13, 32, and 1 neurons). We use the Adam optimizer for parameter updating and set the learning rate $\eta$ of the Actor and Critic models to 0.0005 and 0.001, respectively. We set the discount factor $\gamma$ to 0.99, the GAE parameter $\lambda$ to 0.95, the batch size $N$ of the data sampled for training to 5, the clipping parameter $\epsilon$ to 0.2, and the balance factor $\alpha$ to $0.001 \times \#$ of $episode$. We design comprehensive experiments to answer the following four Research Questions (RQs):

**RQ1 (Effectiveness):** Can FitLight explore a better TSC strategy starting from imitation learning?

**RQ2 (Generalizability):** Can FitLight be plug-and-played cross different traffic environment?

**RQ3 (Benefits):** Why can FitLight improve both learning efficiency and generalizability?

**RQ4 (Applicability):** Can FitLight adapt to real-world situations with very limited resources?

**Baselines.**    For a fair comparison, we chose eight representative baseline methods with constant duration, including three Non-RL methods and five RL-based methods as follows: i) **FixedTime** (Koonce & Rodegerdts, 2008); ii) **MaxPressure** (Varaiya, 2013); iii) **MaxHP**, our proposed expert algorithm; iv) **PressLight** (Wei et al., 2019a); v) **A2C** (Ye et al., 2021); vi) **FedLight** (Ye et al., 2021); vii) **PPO** (Schulman et al., 2017), and; viii) **InitLight** (Ye et al., 2023b). On the other hand, to further evaluate the performance of our approach, we also conduct two state-of-the-art dynamic duration-based methods for comparison: i) **FairLight** (Ye et al., 2022), and ii) **IPDALight** (Zhao et al., 2022). Moreover, to validate the effectiveness of our proposed approach, we compare with four ablation methods as follows: i) **FairLight(c)** (Ye et al., 2022), a constant duration version of FairLight; ii) **IPDALight(c)** (Zhao et al., 2022), a constant duration version of IPDALight; iii) **FitLight(p)**, a pressure-based method that replaces the hybrid pressure in the sate representation and reward design of FitLight with pressure, and; iv) **FitLight(mp)**, the light version of the FitLight model pruning. For FitLight(mp), to meet the extremely resource-constrained requirements of (Xing et al., 2022) (i.e., a microcontroller with merely $16\ KB$ RAM and $32\ KB$ ROM), we set the pruning rate of each layer of the model to 0.2, 0.4, and 0.6, respectively. Under this setting, the memory cost of each PPO agent is only $14.83\ KB$.

**Datasets.**    We consider nine public multi-intersection datasets provided by (Wei et al., 2019c). For all datasets, each intersection of all the road networks has four incoming roads and four outgoing roads, where each road has three lanes, i.e., turning left, going straight, and turning right. The details of the datasets are as follows:

- **Synthetic datasets**: Four synthetic datasets, Syn1-4, contain $1 \times 3$, $2 \times 2$, $3 \times 3$, and $4 \times 4$ intersections, respectively. The vehicle arrival rates are modeled using a Gaussian distribution, with an average rate of 500 vehicles per hour for each entry lane.

- **Real-world datasets**: Five real-world datasets (i.e., Hangzhou1, 2, and Jinan1-3) were collected using cameras deployed in the Gudang sub-district of Hangzhou and the Dongfeng sub-district of Jinan. Each dataset from Hangzhou contains 16 intersections arranged in a $4 \times 4$ grid, while each dataset from Jinan includes 12 intersections arranged in a $3 \times 4$ grid.

### 4.1    RESULTS OF THE CONTROL PERFORMANCE (RQ1)

To answer RQ1, we compared FitLight against the fourteen baseline methods in terms of average travel time, training all RL-based methods using 200 episodes. Table 1 shows experimental results for different control methods. For each dataset, the TSC methods with the best or second-best performance are highlighted in bold. From this table, MaxHP can achieve a shorter average travel time than MaxPressure and competitive results compared with some RL-based methods, especially for larger datasets. These results show the effectiveness of our proposed hybrid pressure and illustrate the reason why we chose MaxHP as the expert algorithm. In this table, FitLight can outperform all constant duration methods. This is because, based on our proposed knowledge sharing mechanism,

Table 1: Comparison of average travel time.

| Type | Method | Average Travel Time (seconds) | | | | | | | | |
|---|---|---|---|---|---|---|---|---|---|---|
| | | Synthetic Dataset | | | | Real-world Dataset | | | | |
| | | Syn1 | Syn2 | Syn3 | Syn4 | Hangzhou1 | Hangzhou2 | Jinan1 | Jinan2 | Jinan3 |
| Non-RL | FixedTime | 380.35 | 453.73 | 534.47 | 606.52 | 525.28 | 537.82 | 444.84 | 378.41 | 403.22 |
| | MaxPressure | 122.68 | 162.32 | 245.26 | 310.37 | 404.67 | 456.11 | 373.76 | 371.24 | 356.30 |
| | MaxHP | 122.98 | 159.72 | 211.72 | 267.38 | 362.34 | 419.82 | 330.00 | 330.04 | 320.77 |
| RL | PressLight | 108.23 | 145.43 | 186.28 | 260.41 | 351.55 | 425.61 | 305.21 | 302.56 | 294.08 |
| | A2C | 117.60 | 133.65 | 236.58 | 433.85 | 339.24 | 416.08 | 375.12 | 322.92 | 288.92 |
| | FedLight | 108.71 | 136.45 | 172.45 | 217.58 | 341.94 | 410.40 | 290.38 | 290.58 | 278.69 |
| | PPO | 105.86 | 138.11 | 204.76 | 314.84 | 358.66 | 421.89 | 321.37 | 304.66 | 286.43 |
| | InitLight | 102.86 | 126.44 | 172.36 | 237.52 | 333.80 | **374.60** | 297.58 | 293.81 | 284.24 |
| | FairLight | 96.49 | 121.67 | 156.55 | 198.52 | 316.28 | 365.20 | 262.55 | 257.38 | 250.69 |
| | IPDALight | **89.66** | **110.00** | **146.34** | **182.72** | **299.37** | 403.18 | 264.31 | 256.89 | 253.41 |
| | FairLight(c) | 98.36 | 132.48 | 193.20 | 228.59 | 314.38 | 383.38 | 281.71 | 277.09 | 270.45 |
| | IPDALight(c) | 97.26 | 121.78 | 161.18 | 205.31 | 310.30 | 392.15 | 272.67 | 270.59 | 263.20 |
| | FitLight(p) | 96.13 | 123.62 | 160.23 | 213.52 | 339.15 | 441.40 | 317.18 | 301.83 | 287.60 |
| | FitLight(mp) | **90.41** | **112.98** | **149.05** | **189.40** | **306.39** | 378.84 | 261.82 | 261.50 | 253.83 |
| | FitLight | 91.06 | 113.57 | 150.83 | 189.72 | 306.44 | 378.00 | **260.84** | 260.96 | 253.64 |

Table 2: Comparison of convergence performance for different TSC methods.

| Method | Average Travel Time (seconds) / Start Episode # of Converge (#) | | | | | | | | |
|---|---|---|---|---|---|---|---|---|---|
| | Synthetic Dataset | | | | Real-world Dataset | | | | |
| | Syn1 | Syn2 | Syn3 | Syn4 | Hangzhou1 | Hangzhou2 | Jinan1 | Jinan2 | Jinan3 |
| PressLight | 487.54 (79) | 532.26 (123) | 759.62 (142) | 781.93 (157) | 472.46 (80) | 521.24 (61) | 541.44 (75) | 512.36 (76) | 543.18 (82) |
| A2C | 871.73 (95) | 1306.89 (165) | 1032.85 (N/A) | 1315.05 (N/A) | 1241.92 (122) | 765.53 (65) | 1298.90 (N/A) | 1207.52 (105) | 1224.99 (133) |
| FedLight | 916.81 (53) | 1175.63 (111) | 1309.26 (133) | 1357.66 (59) | 1009.92 (48) | 807.72 (52) | 1152.66 (47) | 1213.16 (110) | 1229.11 (70) |
| PPO | 873.82 (83) | 1056.83 (165) | 1127.96 (145) | 1297.66 (N/A) | 813.43 (111) | 694.34 (100) | 972.88 (110) | 1031.68 (89) | 869.87 (80) |
| InitialLight | 115.73 (1) | 164.63 (2) | 202.51 (2) | **262.45** (5) | 330.78 (1) | 387.06 (1) | 300.42 (1) | 294.18 (1) | 288.42 (1) |
| FairLight | 530.37 (30) | 686.65 (39) | 876.40 (76) | 979.99 (58) | 517.50 (11) | 512.99 (11) | 703.27 (18) | 677.06 (16) | 588.71 (14) |
| IPDALight | 228.7 (3) | 237.61 (3) | 415.12 (14) | 420.39 (3) | 345.61 (2) | 419.51 (23) | 358.12 (10) | 299.72 (2) | 313.30 (2) |
| FairLight(c) | 609.42 (92) | 774.53 (167) | 1003.88 (N/A) | 1030.18 (N/A) | 721.58 (35) | 588.67 (29) | 757.77 (103) | 691.84 (99) | 687.28 (92) |
| IPDALight(c) | 592.45 (13) | 819.53 (14) | 979.09 (11) | 951.04 (11) | 558.56 (3) | 533.64 (12) | 711.45 (3) | 634.07 (3) | 672.85 (12) |
| FitLight(p) | 173.23 (4) | 275.91 (6) | 475.40 (7) | 630.18 (8) | 369.91 (8) | 423.10 (**1**) | 382.60 (6) | 357.85 (6) | 363.30 (7) |
| FitLight(mp) | 104.74 (**1**) | 136.94 (2) | 183.86 (2) | 310.18 (2) | 330.88 (2) | 392.23 (1) | **281.88** (2) | **282.30** (2) | **274.37** (2) |
| FitLight | **99.21** (**1**) | **125.05** (2) | **163.77** (2) | **215.21** (2) | **321.58** (**1**) | **387.98** (**1**) | **272.97** (2) | **274.22** (**1**) | **267.41** (**1**) |

FitLight enables RL agents to jointly explore the optimal control strategy. On the other hand, due to the full use of duration, dynamic duration methods are significantly better than constant methods. However, our FitLight methods can still achieve a similar level with these two dynamic duration methods, with the biggest gap being only 3.51%. Moreover, compared with the original uncompressed model, the performance decrease of FitLight(mp) is negligible, showing the practicality of our model pruning approach.

## 4.2 RESULTS OF THE CONVERGENCE (RQ2)

To evaluate the efficiency and the generalization ability of FitLight, Figure 3 evaluates the convergence performance of the RL-based methods on the nine datasets. Compared to all baseline methods, our FitLight method achieves the lowest average travel time at the beginning of RL training across all datasets, with significantly fewer fluctuations. This is because our proposed knowledge sharing mechanism enables RL agents to perform effective imitation learning with only very few trajectory samples. In addition, although InitLight uses a pre-trained model that can also perform well initially and converge quickly, our FitLight can consistently achieve the lowest average travel time for all datasets in the long run. This result demonstrates that, due to the better initial solution obtained by imitation learning, the subsequent reinforcement learning process of FitLight enables RL agents to achieve a better final result, confirming the seamless transition between imitation learning and reinforcement learning in our method. Note that, for all the evaluated datasets, FitLight can converge within 2 episodes, significantly faster than most baseline methods. The above facts clearly demonstrate the efficiency and generalizability of our FitLight approach.

To further illustrate the advantages of FitLight, Table 2 provides the detailed convergence information for different RL-based methods, focusing on jumpstart performance (i.e., average travel time in the first episode) and the episode at which the convergence begins, where the best and the second-best results are highlighted in bold. Here, we used the criterion of convergence as described in (Zhao et al., 2022). From this table, we can see that FitLight achieves the best jumpstart performance on all datasets and the fastest convergence on most of them. Due to the pre-trained initial model, InitLight can converge faster on some datasets (e.g., Hangzhou1 and Jinan1-3). However, the gap is only 1 episode, and FitLight can achieve better control performance without any pre-training. Note that, due to model pruning, although the jumpstart performance of the pruned FitLight(mp) model has decreased slightly, it can still converge to a final result similar to the unpruned version. These results demonstrate the plug-and-play capability of our FitLight approach for different traffic scenarios.

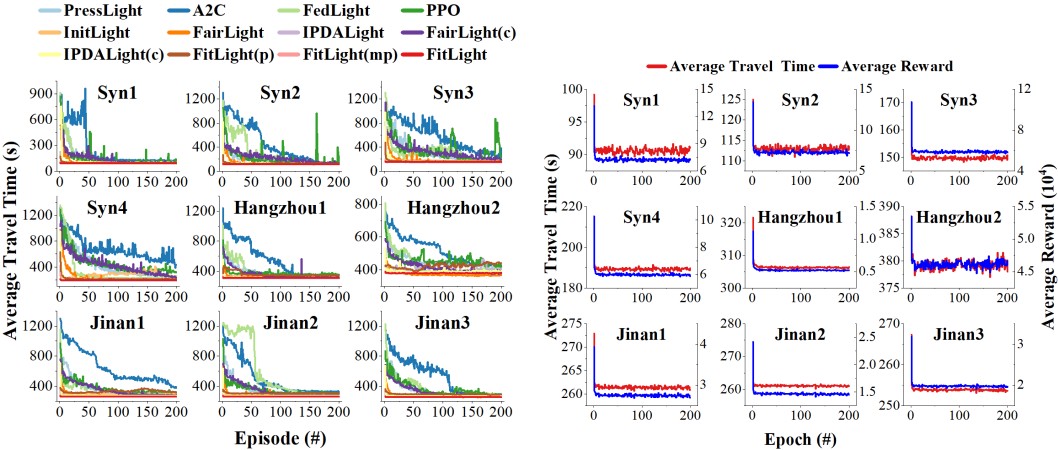

Figure 3: Comparison of convergence rates.    Figure 4: Comparison of travel time and reward.

### 4.3 QUALITY OF THE REWARD FUNCTION (RQ3)

In this paper, we utilize the newly proposed concept of hybrid pressure to represent the state and design the reward of the RL agent, thereby enabling the agent to consider the traffic dynamics of individual vehicles more accurately. To justify our hybrid pressure design and understand why FitLight can be easily applied to different datasets, as shown in Figure 4, we compare the average travel time and average reward of each episode across various datasets. From this figure, we observe that the average travel time is closely correlated with absolute values of the average reward (i.e., the average HP of intersections). These results support the effectiveness of our HP-based agent design in reducing the average travel time of vehicles. Moreover, the smooth change of rewards during training also confirms that our FitLight method supports a smooth transition from imitation learning to reinforcement learning. Thus, our FitLight can improve both the control performance and the generalization ability of RL agents.

### 4.4 RESULTS OF THE DEPLOYMENT COST (RQ4)

To analyze the deployment cost of FitLight, we built a cloud-edge simulation platform consisting of the server mentioned above and 16 Raspberry Pi 4B boards (with an ARM Cortex-A72 CPU and 2GB RAM), on which we deploy a FitLight agent to simulate a real-world intersection scenario. Regarding memory cost, as mentioned above, due to model pruning, the model size of each agent is reduced to only 14.83 KB, which can be deployed on most resource-constrained embedded systems. For computational cost, the agent requires only 0.05ms to make a control decision regarding phase selection. Within an autonomous TSC system, the agent also needs to update its model parameters after collecting some trajectory data. In our approach, an agent takes five samples from its trajectory memory at a time for model update, which costs approximately 51.67ms. Compared with the 10-second signal phase duration, these inference and training costs can meet the real-time requirements of embedded devices in most real-world scenarios. On the other hand, since FitLight includes a federated learning-based knowledge-sharing mechanism, we also evaluate the communication cost of our approach. In the experiment, we set the phase duration of traffic lights to 10 seconds, meaning that every 10 seconds, the agent at an intersection needs to interact with its traffic environment and store a trajectory sample. Once the agent collects a batch of five new samples, it will use these samples to calculate and share the gradient. In other words, every 50 seconds, each edge device needs to send and receive 14.83KB gradients, respectively. This communication cost is tolerable for most IoT devices, as each device incurs a communication overhead of only 2.09MB per hour.

### 5 CONCLUSION

Due to trial-and-error attempts during the training process, existing RL-based TSC methods suffer from the problem of high learning costs and poor generalizability. To address this problem, this paper proposes a novel FIL-based approach named FitLight, which enables RL agents to be plug-and-play for any traffic scenario. Based on the proposed federated imitation learning frameworks and hybrid pressure-based agent design, our FitLight agent can smoothly transition from imitation learning to reinforcement learning. Therefore, the RL agent can quickly find a high-quality initial solution and then find a better final control strategy. Experimental results on various well-known benchmarks show that, compared with the state-of-the-art RL-based TSC methods, FitLight not only converges faster to competitive results but also exhibits stronger robustness in different traffic scenarios. Especially, our approach can achieve near-optimal performance in the first episode.

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
