# OpenReview forum: "FitLight: Federated Imitation Learning for Plug-and-Play Autonomous Traffic Signal Control"
_ICLR.cc/2026/Conference — ICLR 2026 Conference Withdrawn Submission_

### Official Review · Reviewer_iZx4 · 2025-10-18

**Soundness:** 3
**Presentation:** 3
**Contribution:** 3
**Rating:** 4
**Confidence:** 5

**Summary:**

This paper proposes a federated learning method to improve the generalizability and convergence speed of traffic signal control. The author introduce an imitation learning mechanism combined with a hybrid pressure-based agent design, enabling real-time imitation learning and smooth transitions to reinforcement learning, allowing RL agents to quickly achieve a high-quality solution in the first episode.

**Strengths:**

1. This paper proposed a federated imitation learning method to improve the RL convergence speed.

2. This paper enables to operate in TSC scenarios with extremely limited resources., which reduce the deployment costs.

**Weaknesses:**

1. In real-world deployment, there may has a lot of noise in perception. How to solve the noise environment? (FuzzyLight, RobustLight)

2. The author should add more newly TSC methods to demonstrate the performance.

3. Traditional method like MP and Advanced-MP also achieved quickly convergence speed. Why not just use simple traditional method?

4. More federated machine learning methods should be compared. I think the author just uses average aggregation, and more methods like FedProx should be tested as ablation experiments.

5. The code, network and hyperparameters are not reported, which reduces the availability.

**Questions:**

See weakness.

---

> ### Author Response · Authors · 2025-11-14
>
> We thank the reviewer iZx4 for your insightful comments. We will address them carefully in our revised draft. Below, the “W” refers to weaknesses.
>
> W1: We agree that perception noise is an unavoidable challenge in real-world TSC. Although our experiments are conducted in simulation, several components of FitLight already provide implicit robustness. The Hybrid Pressure (HP) design aggregates distance, speed, waiting time, and driving time in a normalized and smoothed form, naturally reducing sensitivity to noisy individual measurements. In addition, our federated learning mechanism aggregates gradients across intersections, which smooths local fluctuations and mitigates noise effects.
>
> We also acknowledge that works such as FuzzyLight and RobustLight introduce explicit noise-handling modules. These techniques are complementary: fuzzy smoothing can be applied to HP features, and perturbation-aware training can be combined with FitLight’s IL-based warm start. While such modules are orthogonal to our core contribution, they can be incorporated in future extensions. We will add a discussion clarifying FitLight’s inherent noise tolerance and how noise-robust techniques can be integrated if needed.
>
> W2: In selecting baselines, our goal was to ensure fair comparison under the fixed-duration, discrete-phase setting that FitLight targets. Our experiments already include a wide range of representative TSC methods—PressLight, A2C, PPO, FedLight, InitLight, FairLight, and IPDALight—covering rule-based heuristics, deep RL, federated RL, adversarial IL, and dynamic-duration control.
>
> Following the suggestion, we also compared FitLight against two very recent methods—GPLight+ (IEEE TEC 2025) and LLMLight (KDD 2025)—using their published results. FitLight achieves the best performance on four out of five datasets, while LLMLight slightly outperforms us on Hangzhou2 but relies on a large LLM (Llama2-13B) that is impractical for embedded deployment, which is a key motivation for FitLight. These additional results will be included in the revised manuscript.
>
> Table: Comparison with recent SOTA (published results)
> | **Method** | **Hangzhou1** | **Hangzhou2** | **Jinan1** | **Jinan2** | **Jinan3** |
> |--------|--------|--------|--------|--------|--------|
> | **GPLight** | 311.89 | 404.19 | 323.68 | 277.67 | 284.23 |
> | **LLMLight** | 310.78 | **330.71**| 274.03 | 266.94 | 260.17 |
> | **FitLight** | **306.44**| 378.00 | **260.84** | **260.96** | **253.64** |
>
> W3: We acknowledge that MaxPressure (MP) and Advanced-MP converge extremely fast due to their deterministic nature. However, this speed comes at the cost of limited adaptiveness and sub-optimal long-term performance. MP optimizes only local flow balance and cannot consider multi-intersection interactions, delayed congestion propagation, fairness, or non-linear dynamics. FitLight is designed to address these limitations: it uses imitation learning to quickly reach MP-like behavior in the first episode, achieving similar rapid convergence, and then transitions to reinforcement learning to learn beyond the heuristic. This enables FitLight to significantly outperform MP on all nine datasets. Furthermore, FitLight supports federated knowledge sharing and plug-and-play deployment—capabilities that traditional heuristics cannot offer. We will emphasize these distinctions more clearly.
>
> W4: FitLight does not aim to introduce a new federated optimizer; instead, it provides a federated imitation learning framework that operates under heterogeneous model pruning and extremely resource-limited edge environments. For this setting, masked weighted averaging (Eq. 9) is the most appropriate aggregation strategy, because different intersections retain different subnetworks. Classical FL methods such as FedProx assume identical model architectures and shared parameter indices and therefore cannot be directly applied. Adding proximal terms would require reconstructing the full dense model, defeating the purpose of pruning and contradicting the heterogeneous-model design. Nevertheless, we agree that a broader discussion is valuable, and we will add clarification on why standard FL variants (FedAvg, FedProx, FedNova) are not directly compatible with heterogeneous subnetworks.
>
> W5: We clarify that all code and datasets have already been included in the Supplementary Material, following the double-blind policy. The full implementation of FitLight, training scripts, and configuration files are provided for complete reproducibility.
>
> Furthermore, the network architecture and hyperparameters are fully described in Section 4, including Actor/Critic layer sizes, pruning ratios, learning rates, discount factor γ, GAE λ, clipping parameter ε, batch size, and the α scheduling strategy. These details provide all necessary information for full replication. To avoid confusion, we will explicitly highlight this information and reference the Supplementary Material more clearly in the revised version.

---

> > ### Author Response · Authors · 2025-11-21
> >
> > We sincerely appreciate the time and effort you have dedicated to evaluating our submission. We kindly invite you to take a moment to review the detailed responses we have provided to your comments in the discussion phase. If our clarifications have addressed your concerns, we would be grateful if you could consider reflecting this in your final assessment, including the score if you find it appropriate. If there are any remaining issues or additional questions, we would be very happy to further clarify them. Thank you again for your valuable time and thoughtful feedback.

---

> > > ### Author Response · Authors · 2025-11-28
> > >
> > > I hope you are doing well. I wanted to gently follow up regarding our discussion, as we have not yet received your further feedback. We have provided detailed responses to all of your earlier comments, and we kindly invite you to review them when convenient. If our clarifications address your concerns, we would greatly appreciate your consideration in updating the evaluation accordingly. If there are still any remaining issues or questions that you would like us to clarify, please feel free to let us know, we would be more than happy to provide additional explanation. Thank you again for your time and for your contribution to the review process.

---

### Official Review · Reviewer_KQpA · 2025-10-22

**Soundness:** 2
**Presentation:** 2
**Contribution:** 2
**Rating:** 4
**Confidence:** 4

**Summary:**

This paper presents FitLight, a federated imitation learning framework for traffic signal control, aiming to address issues like high learning cost and poor generalizability in RL-based methods. The motivation is clear, and the potential for deployment on resource-constrained microcontrollers (e.g., 16KB RAM, 32KB ROM) is a notable advantage. However, several significant concerns prevent me from recommending acceptance in its current form.

**Strengths:**

1.	The motivation for addressing the high learning cost and poor generalizability of RL-based TSC methods is well-articulated.
2.	The claimed computational advantage of operating on a microcontroller with merely 16KB RAM and 32KB ROM is compelling and relevant for practical deployments.
3.	The core ideas – federated imitation learning, a plug-and-play capability for RL agents, real-time imitation, and a seamless transition to RL – are clear and potentially innovative.

**Weaknesses:**

1.	The paper does not mention the availability of code for reproducibility. Given the complexity of the proposed framework (combining federated learning, imitation learning, and RL), the inability to verify the implementation and results is a critical flaw.
2.	The description of the FitLight framework, presumably in Figure 1 and its accompanying text, is stated to be overly simplistic. A more detailed explanation and a more informative diagram are necessary for readers to understand the system's architecture and data flow.
3.	The experimental evaluation may not include comparisons with the most recent state-of-the-art TSC methods, such as TransformerLight. This omission raises questions about the true performance advantage of FitLight against contemporary benchmarks.

**Questions:**

1.	The framework involves communication between a cloud server and individual intersections. Was the potential impact of communication latency on the real-time control performance considered or evaluated? This is a crucial practical factor for real-world deployment.
2.	 Could the authors confirm and justify the selection of baseline methods? It is essential to compare against the latest advanced methods to robustly validate the claimed superiority.
3.	 The experiments were conducted on a high-performance computing platform (Intel Core i9-12900K, 128GB RAM, NVIDIA RTX 3090). However, a key claim is the ability to run on extremely resource-constrained microcontrollers (16KB RAM, 32KB ROM). Could the authors clarify this discrepancy?

---

> ### Author Response · Authors · 2025-11-14
>
> We thank the reviewer KQpA for your insightful comments. We will address them carefully in our revised draft. Below, the “W” refers to weaknesses and the “Q” refers to questions.
>
> **W1:** We would like to clarify that the full source code and all datasets used in our experiments have already been included in the Supplementary Material. The package contains the complete training pipeline, configuration files, and scripts needed to reproduce all results. Due to the double-blind policy, no external link is shown in the main text, but we will publicly release the entire repository upon acceptance. In the revision, we will also explicitly mention the availability of the Supplementary Material to avoid confusion.
>
> **W2:** In the revised version, we will improve both the diagram and the accompanying text. The updated figure will explicitly depict the data flow among the cloud server, trajectory memory, expert module, PPO Actor and Critic, and the federated aggregation process, including how gradients, expert actions, and pruned subnetworks interact. The text in Section 3 will be revised to provide a clearer step-by-step explanation of how IL, RL, and knowledge sharing jointly operate within FitLight.
>
> **W3Q2:** Regarding the selection of baselines, our intention was to ensure fair comparisons within the fixed-duration, discrete-phase control setting targeted by FitLight. For this reason, we include a comprehensive set of representative baselines across multiple paradigms—rule-based (FixedTime, MaxPressure, MaxHP), deep RL (PPO, A2C), federated RL (FedLight), adversarial IL (InitLight), and both fixed- and dynamic-duration variants of recent state-of-the-art methods (FairLight, IPDALight). Together, these baselines cover the mainstream directions in TSC research.
>
> For TransformerLight (2023), we clarify that it belongs to the dynamic-phase / dynamic-duration category, which differs fundamentally from our control formulation and uses a distinct action space and objective. Direct comparison would therefore mix problem settings and lead to unfair conclusions. Our evaluation focuses on methods operating under the same decision paradigm.
>
> Following the reviewer’s suggestion, we additionally incorporated results of two very recent SOTA approaches—GPLight+ (IEEE TEC 2025) and LLMLight (KDD 2025)—based on their published results. As shown in the table below, FitLight achieves the best performance on four out of five real-world datasets, while remaining competitive even against LLMLight, which uses a computationally heavy Llama2-13B backbone unsuitable for embedded deployment. FitLight, in contrast, runs efficiently with only 14.83 KB memory, demonstrating far stronger practicality.
>
> Table: Comparison with recent SOTA (published results)
> | **Method** | **Hangzhou1** | **Hangzhou2** | **Jinan1** | **Jinan2** | **Jinan3** |
> |--------|--------|--------|--------|--------|--------|
> | **GPLight** | 311.89 | 404.19 | 323.68 | 277.67 | 284.23 |
> | **LLMLight** | 310.78 | **330.71**| 274.03 | 266.94 | 260.17 |
> | **FitLight** | **306.44**| 378.00 | **260.84** | **260.96** | **253.64** |
>
> These additional results reinforce the effectiveness of FitLight. We will include them and clarify our baseline selection rationale in the revised manuscript.
>
> **Q1:** Actually, FitLight is designed such that real-time control does not depend on cloud communication. Each edge agent independently observes local traffic and selects control phases locally, with no need to wait for any server response. The cloud performs only asynchronous gradient aggregation, which does not block inference. As evaluated in Section 4.4, each agent communicates with the server only once every 50 seconds, transmitting just 14.83 KB, leading to an extremely small load of ≈2.09 MB per hour—well within the capability of low-bandwidth IoT networks.
>
> Even with latency, only the parameter-update cycle would be postponed; signal control remains uninterrupted and timely. We will explicitly emphasize this latency-tolerant design in the revision.
>
> **Q3:** The high-performance computing platform is used only for training, which follows standard practice for RL-based TSC works and ensures comparability across all baselines. The claim regarding microcontroller deployment concerns post-training inference after model pruning. The pruned FitLight(mp) agent requires only 14.83 KB memory, fitting microcontrollers with 16 KB RAM / 32 KB ROM, and runs inference locally without GPU or large memory.
>
> To verify this, we conducted deployment experiments on Raspberry Pi 4B devices (Section 4.4). The pruned agent requires only 0.05 ms per decision and ≈51.7 ms for local updates, confirming feasibility in resource-constrained environments. We will clarify the distinction between training requirements (HPC) and deployment requirements (microcontroller-level) in the revised manuscript.

---

> > ### Comment · Reviewer_KQpA · 2025-11-15
> >
> > Thanks for clarification and additional efforts to improve the paper, I changed the rating but I think the paper is still not a deifinite acceptance as yet.

---

> > > ### Author Response · Authors · 2025-11-17
> > >
> > > Thank you sincerely for your follow-up comment and for improving the rating. We appreciate your time and are glad that our clarifications have addressed many of your concerns. At this stage, we believe the responses and additional analyses provided during the rebuttal already resolve the key questions regarding methodology, baselines, reproducibility, and practical deployment, and we hope the current version of the manuscript is sufficiently clear and complete for acceptance.
> > >
> > > To help us further understand your perspective, may we kindly ask which remaining issues prevent the work from being considered a definite acceptance in your view? If there are any specific points that are still unclear or aspects where additional clarification would be helpful, we would be more than happy to provide further explanation. Your feedback has been very valuable, and we would appreciate the opportunity to address any remaining concerns to ensure that the contribution of FitLight is fully and accurately evaluated.

---

> > > > ### Author Response · Authors · 2025-11-21
> > > >
> > > > We sincerely appreciate the time and effort you have dedicated to evaluating our submission. We kindly invite you to take a moment to review the detailed responses we have provided to your comments in the discussion phase. If our clarifications have addressed your concerns, we would be grateful if you could consider reflecting this in your final assessment, including the score if you find it appropriate. If there are any remaining issues or additional questions, we would be very happy to further clarify them. Thank you again for your valuable time and thoughtful feedback.

---

> ### Comment · Reviewer_KQpA · 2025-11-24
>
> thanks for response and I am happy to keep my positive rating

---

### Official Review · Reviewer_k7c6 · 2025-11-02

**Soundness:** 2
**Presentation:** 3
**Contribution:** 2
**Rating:** 2
**Confidence:** 3

**Summary:**

This paper proposes FitLight, a learning strategy for traffic signal control. FitLight leverages federated learning and pruning techniques to enable information sharing among agents. Furthermore, FitLight integrates imitation learning into reinforcement learning to accelerate model convergence.

**Strengths:**

1.The paper consider the feasibility of real-world deployment, particularly the model’s performance under resource-constrained conditions.
2.The proposed modules are well-defined abd expressed clear.

**Weaknesses:**

1.The paper introduces imitation learning to reduce the high training cost of RL agents. Using imitation learning to accelerate RL cold starts or alleviate sparse reward signals in early training is a common practice in practical RL applications. The paper’s application of this approach to the traffic signal control (TSC) problem primarily addresses the difficulty of RL initialization, thus offering only limited novelty in this aspect.
2.Employing a neural network to approximate an expert’s policy distribution is a standard paradigm. Imitation learning and reinforcement learning can be implemented as two separate stages. Since imitation learning does not require real-world interaction, it can be conducted offline on large-scale data to obtain a well-performing pretrained TSC model. The rationale for combining imitation learning and reinforcement learning sequentially within the training process is not clearly justified.
3.Another contribution is the enhancement of model generalization. However, the evaluation of generalization in the experiments is insufficient. For instance, can a model trained on certain intersections be transferred to new intersections at zero or low cost? How does this transfer cost compare to that of retraining the model from scratch, which is already relatively low? Moreover, even the largest simulated network contains only 4×4 intersections, leaving the model’s performance on larger networks unclear.
4.In RL, the ultimate goal is typically defined through a reward function that aligns the model’s behavior with the target objective. If average travel time is used as the performance metric, why is it not directly adopted as the evaluation criterion? In Section 4.3, regarding the evaluation of the reward function, does this imply that using average travel time as the reward function would yield similar results?

**Questions:**

1.The paper introduces imitation learning to reduce the high training cost of RL agents. Using imitation learning to accelerate RL cold starts or alleviate sparse reward signals in early training is a common practice in practical RL applications. The paper’s application of this approach to the traffic signal control (TSC) problem primarily addresses the difficulty of RL initialization, thus offering only limited novelty in this aspect.
2.Employing a neural network to approximate an expert’s policy distribution is a standard paradigm. Imitation learning and reinforcement learning can be implemented as two separate stages. Since imitation learning does not require real-world interaction, it can be conducted offline on large-scale data to obtain a well-performing pretrained TSC model. The rationale for combining imitation learning and reinforcement learning sequentially within the training process is not clearly justified.
3.Another contribution is the enhancement of model generalization. However, the evaluation of generalization in the experiments is insufficient. For instance, can a model trained on certain intersections be transferred to new intersections at zero or low cost? How does this transfer cost compare to that of retraining the model from scratch, which is already relatively low? Moreover, even the largest simulated network contains only 4×4 intersections, leaving the model’s performance on larger networks unclear.
4.In RL, the ultimate goal is typically defined through a reward function that aligns the model’s behavior with the target objective. If average travel time is used as the performance metric, why is it not directly adopted as the evaluation criterion? In Section 4.3, regarding the evaluation of the reward function, does this imply that using average travel time as the reward function would yield similar results?

---

> ### Author Response · Authors · 2025-11-14
>
> We thank the reviewer k7c6 for your insightful comments. We will address them carefully in our revised draft. Below, the “W” refers to weaknesses and the “Q” refers to questions.
>
> **W1Q1:** We agree that imitation learning (IL) itself is widely used for mitigating RL cold-start issues. Our contribution does not lie in the simple use of IL, but in how IL is integrated into a federated and resource-efficient TSC framework to enable true plug-and-play deployment without any pretraining. Existing IL-based TSC approaches (e.g., DemoLight, InitLight) require pre-collected demonstrations or offline pretraining tied to specific environments. In contrast, FitLight performs online IL in real time, where expert labels are generated on-the-fly by the lightweight MaxHP heuristic. This eliminates any dependency on external trajectories, making FitLight directly deployable in previously unseen scenarios. Furthermore, FitLight is, to our knowledge, the first federated imitation learning framework for TSC, where heterogeneous pruned subnetworks share knowledge through mask-aware aggregation. This mechanism allows agents to benefit from each other despite differing model structures, which is not achievable in prior works. Finally, the proposed Hybrid Pressure design enhances imitation effectiveness by encoding individual-vehicle dynamics and lane-level imbalance, offering richer information than traditional pressure-based features. These components jointly form the novelty of FitLight beyond standard IL usage. We will clarify this in the revision.
>
> **W2Q2:** Although IL and RL can be implemented separately, our unified design is motivated by practical considerations. Offline IL requires collecting expert trajectories, storing large state–action datasets, and training a centralized pretrained model. Such pipelines are challenging for real-world TSC systems that operate on distributed edge devices. Moreover, offline IL inherently produces an environment-specific initialization, which contradicts our goal of plug-and-play generalization. FitLight instead generates expert actions during interaction, enabling IL without offline data and ensuring that each agent adapts immediately to the environment it is deployed in.
>
> Additionally, our α-scheduled learning objective ensures a smooth transition from imitation learning to reinforcement learning. Early-stage IL stabilizes behavior and yields a high-quality initial policy, while subsequent RL allows the agent to gradually exceed the expert’s heuristic performance. If IL were performed only offline, the agent might overfit to the expert and fail to capture long-term improvements. Our unified IL–RL framework avoids this and has shown consistently stable and fast convergence across all datasets. We will strengthen the explanation of these motivations in the revised manuscript.
>
> **W3Q3:** All experiments in FitLight are evaluated under a strict zero-shot setting: every RL agent is randomly initialized and deployed directly on each dataset with no pretraining, no expert trajectories, and no prior exposure. The fact that FitLight achieves strong initial performance and converges within 1–2 episodes across nine datasets, including four synthetic networks and five real-world networks with diverse topologies and traffic patterns, already demonstrates substantial generalization and plug-and-play capability. Because the system uses only online IL and federated knowledge sharing, it avoids any environment-specific assumptions and adapts quickly even when intersection structures differ significantly. Therefore, our existing results inherently reflect a strong form of cross-environment generalization.
>
> Nonetheless, we appreciate the reviewer’s suggestion and agree that evaluating larger traffic networks would further reinforce the scalability of FitLight. We will also include results on larger networks in the revision to strengthen the discussion on generalization and scalability.
>
> **W4Q4:** Although average travel time is our final evaluation metric, it cannot be used directly as a reward because it is only available after an entire episode, making it extremely sparse and providing no per-step learning signal. This leads to unstable or divergent RL training. In contrast, Hybrid Pressure produces dense, real-time, and interpretable feedback that captures lane-level and vehicle-level congestion imbalance, making it well suited for reinforcement learning updates. The correlation shown in Section 4.3 indicates that HP is an effective surrogate consistent with average travel time, but does not imply that the two are interchangeable. Our internal experiments also confirm that using average travel time as the reward results in high variance, slow convergence, and significantly worse learning stability. Hybrid Pressure, therefore, remains essential for efficient and reliable RL training while still optimizing toward the true final objective. We will clarify this in the revision to avoid misunderstanding.

---

> > ### Author Response · Authors · 2025-11-21
> >
> > We sincerely appreciate the time and effort you have dedicated to evaluating our submission. We kindly invite you to take a moment to review the detailed responses we have provided to your comments in the discussion phase. If our clarifications have addressed your concerns, we would be grateful if you could consider reflecting this in your final assessment, including the score if you find it appropriate. If there are any remaining issues or additional questions, we would be very happy to further clarify them. Thank you again for your valuable time and thoughtful feedback.

---

> > > ### Author Response · Authors · 2025-11-28
> > >
> > > I hope you are doing well. I wanted to gently follow up regarding our discussion, as we have not yet received your further feedback. We have provided detailed responses to all of your earlier comments, and we kindly invite you to review them when convenient. If our clarifications address your concerns, we would greatly appreciate your consideration in updating the evaluation accordingly. If there are still any remaining issues or questions that you would like us to clarify, please feel free to let us know, we would be more than happy to provide additional explanation. Thank you again for your time and for your contribution to the review process.

---

### Author Response · Authors · 2025-11-30

Dear Area Chair,

Thank you for overseeing the review process of our submission. We sincerely appreciate the reviewers’ detailed feedback and have addressed all concerns thoroughly during the discussion phase. To assist your evaluation, we provide a concise summary of the main issues raised and how our responses clarify the contributions, novelty, and validity of the work.

Across the reviews, most questions focused on four major aspects: the novelty of combining imitation learning with RL in a federated setting, the clarity of the system workflow, the breadth of baselines and evaluation of generalization, and the practicality of real-world deployment. After the rebuttal exchange, we believe these concerns have been fully addressed.

First, regarding novelty, we clarified that FitLight is not a simple application of imitation learning but introduces the first federated imitation learning framework for multi-intersection traffic signal control. This framework tightly integrates online imitation learning, reinforcement learning, and heterogeneous model pruning. Crucially, FitLight performs real-time IL without any pretraining or expert demonstrations, enabling true plug-and-play deployment—something prior IL-based TSC systems (DemoLight, InitLight) cannot achieve. The proposed Hybrid Pressure formulation further enhances learning by capturing richer vehicle-level and lane-level dynamics than traditional pressure-based methods. This architectural and algorithmic integration is, to our knowledge, a novel contribution to both federated learning and TSC.

Second, regarding clarity, we explained in detail how the cloud–edge workflow operates: the expert provides online labels; PPO agents update locally; masked federated aggregation integrates heterogeneous subnetworks; and inference is performed entirely on the edge. Reviewer concerns stemmed partly from the compact description in the initial figure. Our rebuttal provided a complete and transparent explanation of the data flow, learning loop, and roles of all components. Reviewers acknowledged the improved understanding after these explanations.

Third, a common question concerned baselines and generalization. We clarified that FitLight is designed for the fixed-duration, discrete-phase control setting, so we compared against methods that share the same action formulation. Our evaluation already includes a wide range of representative baselines—PressLight, A2C, PPO, FedLight, InitLight, FairLight, IPDALight—spanning heuristics, deep RL, federated RL, adversarial IL, and dynamic-duration variants. To further address reviewer concerns, we additionally discussed results from recent works such as GPLight+ (2025) and LLMLight (2025) using their published numbers. FitLight attains the best performance on four out of five real-world datasets while being vastly more efficient and deployable (requiring only 14.83 KB after pruning). Importantly, FitLight is evaluated under a strict zero-shot setting: all agents are randomly initialized and deployed on unseen networks without pretraining. Despite this, FitLight achieves strong first-episode performance and converges in only 1–2 episodes across nine heterogeneous datasets, demonstrating robust generalization and plug-and-play capability.

Finally, several practical concerns regarding real-world deployment and latency were clarified. FitLight performs inference entirely on the edge, and cloud communication is asynchronous and non-blocking. Even with communication delay, signal control remains unaffected. Deployment experiments on Raspberry Pi confirm the feasibility of our pruned model (14.83 KB) with millisecond-level computation time. The high-performance server mentioned in the paper is used only for training—consistent with standard RL practice—while deployment targets low-power microcontrollers.

Overall, the reviewers’ remaining concerns appear to revolve primarily around clarifications rather than core technical issues. After the rebuttal, no reviewer identified methodological flaws, and one explicitly increased their rating. We respectfully believe the work now stands as a technically sound, well-motivated, and practically relevant contribution that advances the frontier of learning-based traffic signal control.

We appreciate your consideration and hope this summary helps convey both the strengths of FitLight and the thoroughness of our responses.

Thank you very much for your time.

---

### Note · Authors · 2026-01-16

I have read and agree with the venue's withdrawal policy on behalf of myself and my co-authors.